# Contribution of physician assistants/associates to secondary care: a systematic review

Mary Halter,[1] Carly Wheeler,[1] Ferruccio Pelone,[2] Heather Gage,[3] Simon de Lusignan,[4] Jim Parle,[5] Robert Grant,[1] Jonathan Gabe,[6] Laura Nice,[5] Vari M Drennan[1]

[1]Faculty of Health, Social Care and Education, Kingston University and St George's, University of London, London, UK
[2]National Guideline Alliance, Royal College of Obstetricians and Gynaecologists, London, UK
[3]School of Economics, University of Surrey, Guildford, Surrey, UK
[4]Department of Clinical and Experimental Medicine, University of Surrey, Guildford, Surrey, UK
[5]Institute of Clinical Sciences, University of Birmingham, Birmingham, UK
[6]Centre for Criminology and Sociology, School of Law, Royal Holloway, University of London, London, UK

**Correspondence to**
Mary Halter;
m.halter@sgul.kingston.ac.uk

## ABSTRACT

**Objective** To appraise and synthesise research on the impact of physician assistants/associates (PA) in secondary care, specifically acute internal medicine, care of the elderly, emergency medicine, trauma and orthopaedics, and mental health.

**Design** Systematic review.

**Setting** Electronic databases (Medline, Embase, ASSIA, CINAHL, SCOPUS, PsycINFO, Social Policy and Practice, EconLit and Cochrane), reference lists and related articles.

**Included articles** Peer-reviewed articles of any study design, published in English, 1995–2017.

**Interventions** Blinded parallel processes were used to screen abstracts and full text, data extractions and quality assessments against published guidelines. A narrative synthesis was undertaken.

**Outcome measures** Impact on: patients' experiences and outcomes, service organisation, working practices, other professional groups and costs.

**Results** 5472 references were identified and 161 read in full; 16 were included—emergency medicine (7), trauma and orthopaedics (6), acute internal medicine (2), mental health (1) and care of the elderly (0). All studies were observational, with variable methodological quality. In emergency medicine and in trauma and orthopaedics, when PAs are added to teams, reduced waiting and process times, lower charges, equivalent readmission rate and good acceptability to staff and patients are reported. Analgesia prescribing, operative complications and mortality outcomes were variable. In internal medicine outcomes of care provided by PAs and doctors were equivalent.

**Conclusions** PAs have been deployed to increase the capacity of a team, enabling gains in waiting time, throughput, continuity and medical cover. When PAs were compared with medical staff, reassuringly there was little or no negative effect on health outcomes or cost. The difficulty of attributing cause and effect in complex systems where work is organised in teams is highlighted. Further rigorous evaluation is required to address the complexity of the PA role, reporting on more than one setting, and including comparison between PAs and roles for which they are substituting.

**PROSPERO registration number** CRD42016032895.

## INTRODUCTION

Healthcare systems internationally face substantial medical workforce challenges.[1] An approach used in many countries has been to develop advanced clinical practitioner roles (also sometimes known as mid-level non-physician clinicians), who undertake some of the activities of doctors.[2] One of these roles is the physician assistant/associate (PA). The PA role was first developed by physicians in the 1960s in the USA in response to medical shortages in certain specialties and regions.[3] As of the end of 2016, there were 115 547 nationally certified and state-licensed PAs in the USA,[4] following 44% growth since 2010. In the USA, PAs practice as medical professionals in healthcare teams with physicians and other providers in all 50 states.[5] Over the last two decades other countries have been introducing PAs into their health workforce, including Australia, Canada, Germany, Ghana, India, Kenya, the Netherlands, Saudi Arabia, South Africa, Taiwan and the UK,[6] where they are known as physician associates. Some countries, including the UK, have national or federal policy commitments

to develop PA education programmes and significantly increase their availability,[7 8] while others are determining the value of such roles through demonstration projects.[9] The role has received increasing attention as a potential growth area from the UK government, particularly in primary care[10] where there is evidence that PAs can be complementary to general practitioner (GP) and nursing roles, although with limitations due to not currently having prescribing rights.[11] However, in the USA only 21% of PAs work in family medicine/general practice[4]; similarly in the UK and the Netherlands they report working in a range of secondary care specialties.[12 13]

► This study's strengths lie in systematically analysing the empirical evidence for the contribution of physician associates (PA) to secondary care, following international guidelines.

► Focusing on specialties in which PAs are increasingly deployed in the UK, while aiming for international applicability. This methodological approach carries limitations in excluding closely related and sometimes high-quality studies that did not meet our strict inclusion criteria, but that are relevant to understanding the impact of PAs in secondary care settings.

► The review was strengthened by using established guidelines to carry out quality assessment of the included studies. Although our approach can be considered reductionist, it provides decision makers with consistent information about the quality of the evidence against which to weight the value of individual findings.

Like many aspects of workforce innovation and change, there is very limited published evidence as to the contribution and impact PAs have within this setting. Existing systematic reviews of the contribution PAs make to healthcare have considered evidence from primary and secondary care together,[14] just primary care,[15] rural healthcare and emergency department (ED)[16] or considered PAs and nurse practitioners together in surgical services.[17] Given the recent trends to use PAs internationally in secondary care, our purpose in conducting this new review was to systematically summarise the current evidence in secondary care.

The objective of the review was to appraise and synthesise the published literature on the impact of PAs on patient experience and outcomes, service organisation, working practices, other professional groups and cost. The review was bounded by consideration of the secondary care specialties in which PAs were most frequently reported to be employed in the UK. Using the annual UK Association of Physician Associates Census (conducted in 2016 with 150 PA respondents),[18] four specialties with relatively larger numbers of PAs replying to the survey were clearly identifiable: acute internal medicine (n=23), emergency medicine (n=23), care of the elderly (n=12) and trauma and orthopaedics (n=10). While three other specialties (cardiology, neurology and general surgery) reported five PAs in each, we selected mental health as our fifth specialty to explore, with four PAs reported,[18] to provide a contrast to the focus on physical health in the other four specialties selected. The concentration of PAs in these clinical areas is consistent with evidence from other European countries developing a PA workforce.[19] The review is intended to inform clinicians and managers considering innovation and change in their secondary care workforce.

## METHODS
### Search strategy
This systematic review was designed and reported to meet international guidelines: the Preferred Reporting Items for Systematic Reviews and Meta-Analyses (PRISMA).[20] Full details of the overall search strategy can be found in the research protocol, registered with the International Prospective Register of Systematic Reviews (PROSPERO), CRD42016032895.[21]

Studies addressing the research question were identified by systematic searching for keywords in the following electronic databases: Medline (Ovid), Embase (Ovid), Applied Social Sciences Index and Abstracts (ASSIA), Cumulative Index to Nursing and Allied Health Literature (CINAHL) Plus (EBSCO), SCOPUS—V.4 (Elsevier), PsycINFO, Social Policy and Practice (Ovid), EconLit (EBSCO) and Cochrane Central Register of Controlled Trials (CENTRAL) from the beginning of January 1995 to the beginning of January 2018. The search strategy was performed on 14 December 2015 and updated on 5 January 2018. No language or publication status restrictions were imposed at the electronic search strategy stage. We present the Medline search strategy, and the definitions of the MeSH terms employed, in online supplementary file 1.

In addition, we used 'lateral searching' techniques[22] including checking reference lists of systematic reviews identified at the abstract screening stage and papers selected for inclusion after full-text reading; using the 'Cited by' option on Scopus, and the 'Related articles' option on PubMed, and tracking citations.

### Inclusion criteria and study selection
Relevant studies were selected according to eligibility criteria using a two-step screening process: (1) title and abstract screening and (2) full-text screening. First, two authors (CW and FP) in parallel sifted titles and abstracts of all the articles resulting from the searches to ascertain their potential relevance, with disagreements resolved by a third author (MH or VMD). All the full texts of the potentially relevant citations were further examined in parallel by two authors (pairings among CW, FP or MH) to analyse whether they met all the inclusion criteria. Disagreements were resolved by peer discussion and a third view from the project lead (VMD) if required.

Peer-reviewed articles were considered for analysis if they fitted the following inclusion criteria:
- Population: PAs according to the UK definition.[23]
- Intervention: the implementation of PAs in the following secondary healthcare specialties: acute medicine, care of the elderly, emergency medicine, mental health, and trauma and orthopaedics (see online supplementary file 2 for the definitions used).
- Comparison: the comparison group was any healthcare professional to whom PAs were compared.
- Outcome: any measure of impact, informed by recognised dimensions of quality—effectiveness, efficiency, acceptability, access, equity and relevance.[24]
- Study design: any study design that allowed measurement of impact of PAs in secondary care utilising a primary study.

### Screening exclusion criteria
Articles were excluded if they did not fulfil one or more inclusion criteria or if they: (1) were not published in the English language; (2) reported on PAs working in countries that are not defined by the International Monetary

Fund as advanced economies[25]; (3) did not report empirical findings or were published only in abstract form; (4) presented their results for PAs in an amalgamated form with the results for other professions/mid-level providers or did not describe the specialties they were reporting on; (5) contained only descriptive accounts of PA demography, workload, clinical practice or productivity or PA self-report of any aspect of their role; (6) focused on and measured an intervention delivered by PAs rather than PAs as the intervention; (7) focused on and measured PA clinical practice or productivity before and after a service redesign or educational intervention; (8) focused solely on educational processes; and (9) presented literature reviews, commentaries and/or non-peer-reviewed articles.

### Data collection and quality assessment

Two authors (pairings among FP, CW and MH) independently extracted the data from selected papers, with any disagreement resolved through discussion. A checklist was used to extract the following information from the selected papers: (1) general characteristics of studies and (2) results, limitations and conclusions as noted by authors and reviewers.

The same author pairings appraised the quality of included studies using the QualSyst quality checklists for quantitative and qualitative studies, selected as a validated tool for the evaluation of primary research papers from a variety of fields,[26] with additional questions from the Mixed Methods Appraisal Tool, selected as a tool tested for its efficiency and reliability,[27] where appropriate. For the quantitative studies, 12 items (figure 1) were scored depending on the degree to which the specific criteria were met ('yes'=2, 'partial'=1, 'no'=0). Scores for the qualitative studies were calculated in a similar fashion, based on the scoring of 10 items. Any items not applicable to a particular study design were marked 'n/a' and were excluded from the calculation of the summary score. No study was excluded on the basis of its quality score; the limitations of lower quality evidence are however explored in considering how much weight can be given to the evidence when we synthesise studies.[28]

### Data analysis

A meta-analysis was not performed due to the heterogeneity of the included studies in terms of scope and outcomes investigated as found during data extraction. Therefore, narrative synthesis was undertaken[29] conducted against the four elements in published, accepted guidance on the conduct of narrative synthesis in systematic reviews[30 31]: developing a theory of how the intervention works, why and for whom; developing a preliminary synthesis of findings of included studies; exploring relationships within and between studies; assessing the robustness of the synthesis (through formal quality assessment as well as reflection). For the synthesis the included studies were grouped into specialty (ie, acute medicine, care of the elderly, emergency medicine, mental health, and trauma and orthopaedics) and then subgrouped into the outcomes they measured.

## RESULTS
### Search results

The overall search strategy identified 5472 references, from which we selected 161 articles for more detailed reading. Figure 2 presents the PRISMA flow chart, illustrating the literature search and selection process, and reasons for study exclusion on full-text reading. A total of 16 articles were included for data collection, quality appraisal and data analysis.

A summary of the included evidence is presented below in three subsections: characteristics of included studies, methodological quality and synthesis of findings on the impact of PAs.

### Characteristics of included studies

Table 1 presents the characteristics for each study in terms of the specialties they were drawn from.

In summary, seven studies were included from emergency medicine,[32–38] six studies reported from trauma and orthopaedics,[39–44] two from acute internal medicine[44–46] and one from mental health.[47] No studies were identified from care of the elderly medicine.

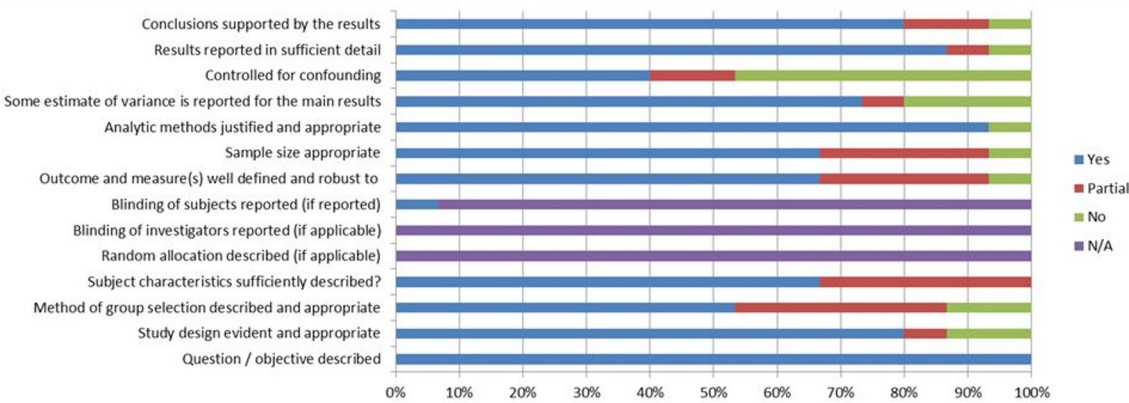

**Figure 1** 'Risk of bias' graph: review authors' judgements about each risk of bias item presented as percentages across all included studies.

**Table 1** Characteristics of studies included in full—studies presenting comparisons of PAs with other healthcare professionals

| Specialty | Aim(s) | Study setting | Intervention | Comparison | Participants | Study design | Outcome measures | First author and year |
|---|---|---|---|---|---|---|---|---|
| Emergency medicine | To determine whether PAs are an appropriate option for providing services rendered by physicians in the ED | USA Walk in urgent care facility (satellite of an inner-city teaching hospital level 1 trauma centre) | PAs (n=5) rotate through the ED. PAs work solo from 08:00 to 12:00. No written diagnostic or therapeutic guidelines were followed. | 25 physicians rotate through the ED. Physicians work solo from 17:00 to 21:00. No written diagnostic or therapeutic guidelines were followed. | n=5345 (seen by PAs) n=4256 (seen by physicians) during times of single coverage June 1995 to June 1996 | Comparative retrospective | ▲ Length of visit ▲ Total charge | Arnopolin (2000)[32] |
| Emergency medicine | To examine the impact of PAs and nurse practitioners in EDs | Canada Six community hospitals with ED volumes between 23 and 66 000 | PAs were introduced as an unregulated provider without medical directives and worked under the supervision of a registered physician who was responsible for all patient care on predetermined busiest periods for each ED. | Baseline 2 weeks | All ED patients: baseline n=9585; 2-week period 6 months postimplementation June 2007 n=10 007, of which PAs were on duty for 1076 visits and directly involved in n=376 | Descriptive retrospective | ▲ Leaving without being seen ▲ Wait time (triage to initial assessment) ▲ Length of stay in ED | Ducharme (2009)[33] |
| Emergency medicine | To understand trends in emergency medicine and interprofessional roles in delivering this care [...] The focus was on how doctors, PAs and nurse practitioners share emergency medicine visits. | USA National sample EDs of non-institutional general and short-stay hospitals in the 50 states and the District of Columbia from the National Hospital Ambulatory Medical Care Survey | PAs as providers of ED care and prescribers of medication in emergency medicine (7.9% of patients seen by PAs in 2004). | Physicians and nurse practitioners | Random sample of patient visits to hospital EDs (n=1 034 758 313), 1995–2004 | Longitudinal | ▲ Proportion of visits in which medications are prescribed ▲ Mean number of prescriptions written per visit ▲ Non-narcotic analgesic prescriptions ▲ Narcotic analgesic/NSAID prescription by type of provider ▲ Patient contact growth by provider | Hooker (2008)[34] |
| Emergency medicine | To compare the analgesic practices of emergency physicians with that of PAs | USA ED within a suburban teaching hospital in Michigan with 90 000 annual visits | PAs were deployed for seeing patients presenting at the ED with isolated lower extremity trauma. PAs work closely with emergency physicians in the Prompt Care Area of the ED. | Emergency physicians | n=384 survey respondents of patients of all ages who presented at the ED with an isolated lower extremity injury evaluated with a foot or ankle radiograph, n=227 PA patients, n=153 emergency physician patients in a 9-week period | Prospective cohort | ▲ Analgesia prescribing | Kozlowski (2002)[35] |
| Emergency medicine | To evaluate PAs' management of paediatric patients in a general ED through examination of the 72 hours' recidivism rates of their younger paediatric patients | USA General urban ED treating approximately 58 000 patients annually, 20% of which are under 18 years | PAs evaluate, treat and discharge patients of any age independent of emergency physicians and PAs treating patients with consult from the emergency physician. | Attending emergency physician only | n=2798 PA only cases; n=984 PA with emergency physician; n=6587 emergency physician only | Comparative retrospective | ▲ 72-hour revisits to the ED | Pavlik (2017)[36] |

Continued

**Table 1** Continued

| Specialty | Aim(s) | Study setting | Intervention | Comparison | Participants | Study design | Outcome measures | First author and year |
|---|---|---|---|---|---|---|---|---|
| Emergency medicine | To compare the quality of ED pain management before and after implementation of the Joint Commission on the Accreditation of Healthcare Organizations' standards in 2001 | USA National sample EDs included in the National Hospital Ambulatory Medical Care Survey | The use of PAs in the care of patients presenting to the ED with a long bone fracture. | Patients presenting to the ED with a long bone fracture not seen by PAs (medical residents, internists) | n=2064 Patients presenting at the ED with a long bone fracture (femur, humerus, tibia, fibula, radius, or ulna) in two time periods: 1998–2000, n=834 of which 3% were seen by a PA, 9% by resident/intern and 90% by staff physician; 2001–2003 8% PA, 10% resident/intern, 90% staff physician | Retrospective cohort | ▲ Proportion of patients with long bone fracture receiving analgesia | Ritsema (2007)[37] |
| Emergency medicine | To compare the wound care practices and infection rates of wounds managed in the ED by practitioners with varying levels of medical training | USA Department of Emergency Medicine within a teaching hospital in New York | All patients with lacerations were evaluated by an attending physician who determined whether wound could be managed by a junior practitioner (PAs, students, interns, and residents). | ED patients whose wounds were managed by other providers (students, interns and residents) | All patients with lacerations attending the ED n=1163, n=901 seen by a PA, n=262 by other providers October 1992 to November 1993 | Prospective observational | ▲ Patient wound infection rate | Singer (1995)[38] |
| Trauma and orthopaedics | To define the clinical and financial impact of hospital-based PAs on orthopaedic trauma care at a level II community hospital | USA Orthopaedic trauma care at a level II community hospital | Hospital-employed PAs (n=2) were used to cover all orthopaedic trauma needs, under the supervision of one of 18 orthopaedic surgeons. Each PA performed 12-hour day shifts for 3 consecutive days, January to December 2007. PAs on call carried trauma pagers and reported to the emergency room as soon as possible. | Attending surgeon as the primary orthopaedic responder for emergency department consults | n=1104 ▲ n=310: PA ▲ n=687: no PA | Comparative retrospective | ▲ Triage time to time seen by orthopaedic service in emergency department (min) ▲ Triage time to time of surgery (min) ▲ Operating room complication rates (%) ▲ The use of deep vein thrombosis prophylaxis (%) ▲ Postoperative antibiotic administration (%) ▲ Postoperative complications (%) ▲ Triage time to out of emergency department (min) ▲ Operating room set-up time (min) ▲ Average operating room time (min) ▲ Time from wound closure to wheels out (operating room) (min) ▲ Hospital length of stay (min) ▲ Cost savings (emergency department) ($) ▲ Cost savings (operating room) ($) | Althausen (2013)[39] |

Continued

**Table 1** Continued

| Specialty | First author and year | Aim(s) | Study setting | Intervention | Comparison | Participants | Study design | Outcome measures |
|---|---|---|---|---|---|---|---|---|
| Trauma and orthopaedics | Bohm (2010)[40] | To describe the effect of PAs working in an arthroplasty practice from the perspective of patients and healthcare providers. To describe the costs, time savings for surgeons and effects on surgical throughput and waiting times | Canada High-volume academic arthroplasty programme employing PAs (The Concordia Joint Replacement Group) | Addition of PAs (n=3) to the operating room team. The PAs were added to the team, replacing surgical assists (usually general practitioners). The PAs took first call with their supervising physician, provided first-assist services in the operating room (OR), write postoperative tests/ investigations, generate operative notes, undertake daily working rounds and complete discharge summaries. | • Costs: GP first assists in the operating room • Waiting times: patients on the arthroplasty waiting list in 2004 and 2005 | Sample size varying by outcome: • Patient satisfaction n=1070 • Perceptions of healthcare providers and patients n=44 • Costs n=402 surgical procedures performed in 2006 • Time savings n=1409 procedures carried out 2006 • Waiting times in 2006 | Mixed methods | ▲ Patient satisfaction ▲ Perceptions of PAs among healthcare providers and patients ▲ Costs ▲ Time savings ▲ Waiting times ▲ Throughput |
| Trauma and orthopaedics | Garrison (2017)[41] | To assess whether the type of provider (attending physician vs PA) or number of providers involved in the non-operative management of a paediatric forearm fracture influenced the risk of that fracture healing as a malunion | USA Children's hospital medical centre | PAs carrying out non-operative management of forearm fractures at orthopaedic clinic visits. | Attending physician | Patient charts of those aged 3–17 years seen at the orthopaedics department February 2012 to January 2013 n=141 | Comparative retrospective | ▲ Fracture malunion (maximum angulation criteria) at last clinic visit |
| Trauma and orthopaedics | Hepp (2017)[42] | To describe the role of the PA in the upper extremity surgical programme; describe the role of the PA in an operating room study; and show the impact of the PA role on patients, providers and the system | Canada Subspecialised upper extremity surgical programme at a peripheral hospital, as part of a Physician Assistant Demonstration project where 12 PAs were introduced to various healthcare settings | One PA filling provider gaps in four areas: preoperative patient screening, assisting in operating room care (including a double-room experiment), aiding in aftercare of surgery and attending to postdischarge follow-up care. | Preoperative—surgeon working alone; operating room—team with surgical assistant or role unfilled and single operating room; surgery aftercare—replacing a postunfilled surgical extender; postdischarge—surgeon only | n=38 interviews; n=75 surveys (n=28 from healthcare providers and 47 from patients) | Mixed methods | ▲ Perceptions and experiences with the PA ▲ Patient rating of quality of care ▲ Expected and actual operating room times ▲ Total new patients seen |
| Trauma and orthopaedics | Mains (2009)[43] | To assess whether staffing changes within a level 1 trauma centre improved mortality and shortened hospital and ICU length of stay for patients with trauma | USA Urban, community-based level 1 trauma centre | Core trauma panel (consisting of full-time, in-house trauma surgeons) and PAs | Group 1: general surgery residents (staffed by full-time, in-house postgraduate year 4 general surgery residents with attending back-up from home, followed by a transition to a trauma service staffed with in-house independent general surgeon attendings); group 2: core trauma panel (consisting of full-time, in-house trauma surgeons, without PAs or residents) | n=15 297 Trauma patients 18 years or older and not transferred from the ED to another acute care facility | Prospective cohort | ▲ Overall mortality ▲ Mortality for patients with injury severity score (ISS)>15 ▲ Hospital LOS |

Continued

**Table 1** Continued

| Specialty | Aim(s) | Study setting | Intervention | Comparison | Participants | Study design | Outcome measures | First author and year |
|---|---|---|---|---|---|---|---|---|
| Trauma and orthopaedics | To analyse patient outcomes and efficiency of care provided for trauma patients during transition from resident physician support to PA support | USA Level I trauma centre | PAs substituting for doctors in trauma alerts: PAs' role was to assist the trauma surgeon at trauma alerts and trauma patient rounds, update the trauma patient census list. | General and orthopaedic residents who attend in trauma alerts | n=293 before n=476-after All patients evaluated by the trauma surgeons and on the trauma registry, excluding those transferred to another facility for treatment of severe burns | Before-after | ▲ Collaborative relationship ▲ Transfer time ▲ LOS ▲ Mortality rate | Oswanski (2004)[44] |
| Internal medicine | To compare outcomes directly from the expanded use of PAs with those of a hospitalist group staffed with a greater proportion of attending physicians at the same hospital during the same time | USA Community hospital with 26 000 adult patients discharged annually | Expanded PA group: used three physicians and three PAs daily for ward rounds with PAs expected to see 14 patients daily, plus one more PA responsible for day shift admissions. PAs worked in dyads with ward round physician; PAs discussed the treatment plans at least once a day with the physician to a written protocol for PA–physician dyad expectations. | Conventional group: used nine physicians and two PAs for rounding, with PAs expected to see nine patients daily, plus day shift admissions by the physician. PAs worked in dyads with ward round physician; PAs discussed the treatment plans at least once a day with the physician. No written protocol for PA–physician dyad expectations. | Patients discharged between January 2012 and June 2013; n=6612 expanded PA group and n=10 352 in the conventional group | Retrospective comparative | ▲ 30-day all-cause readmission ▲ Inpatient mortality ▲ Cost of care ▲ Consultant/attending use ▲ Length of stay | Capstack (2016)[45] |
| Internal medicine | To examine and compare costs, between a PA service and an intern/ resident (teaching) service in the provision of inpatient care for five high-volume internal medicine diagnostic-related groups | USA Two general internal medicine units, teaching hospital | The use of PAs (n=16) in the provision of care within internal medicine department (64 attending physicians on rotation coverage, scheduled to admit to either a PA or teaching service, with group assignment determined 1 year in advance). | The teaching service (32 interns/residents with an average experience of 1-year postmedical school) | Adult patients discharged in the following diagnostic-related groups: cerebrovascular accident/stroke, pneumonia, acute myocardial infarction discharged alive, congestive heart failure, gastrointestinal haemorrhage: n=923, of which n=409 PA and n=514 teaching service | Prospective cohort study | ▲ Relative value units (costs) ▲ Length of stay | Van Rhee (2002)[46] |
| Mental health | To examine the role of PAs in the care of patients with severe and persistent mental illness | Canada Assertive community treatment team, providing multidisciplinary care to patients with severe and persistent mental illness | A PA was hired to assist with intake psychiatric assessments, physical examinations, preventive care, and follow-up of psychiatric and medical complaints in a model of PA supervised by a psychiatrist. | No comparison | Assertive community treatment team members (three social workers, one psychiatrist, two psychiatric nurses, one occupational therapist, one recreational therapist, the PA) | Qualitative interview | ▲ Perceived effect and challenges of delivering psychiatric care with the PA model | McCutchen (2017)[47] |

ED, emergency department; ICU, intensive care unit; LOS, length of stay; NSAID, non-steroidal anti-inflammatory drug; PA, physician assistant/associate.

The publication year ranged from 1995[38] to 2017.[36 41 42 47] The majority were from the USA (n=12), with four from Canada.[32 38 42 47] The studies measured a number of outcomes; results are shown in table 2.

Two studies employed mixed methods[40 42]; one study used a qualitative analysis,[47] the remainder employed quantitative approaches. Five quantitative studies analysed prospectively collected data[35 38 40 43 46] and seven used a retrospective analysis.[32–34 36 39 42 45] All studies but one[46] were observational.

## Methodological quality
The studies were of variable methodological quality. The mean quality score was 79% (SD 0.20), median 82%, minimum 32%,[40] maximum 100%,[37 43] IQR 73, 92. Figure 1 presents a summary of the degree to which the included evidence met the criteria of methodological quality and shows that the most important methodological flaws in the included quantitative studies were the failure to adjust the analysis for confounding variables, the absence of information to evaluate participants' selection adequacy, and the lack of information about baseline and/or demographic information of the investigated participants. Overall, the quality of the included qualitative evidence was low, mainly due to insufficient description of the sampling strategy, data collection and analysis methods.[40 44 47]

## Synthesis of findings on the impact of PAs
We organised our findings by secondary care specialty. Within each specialty, we described the findings within the quality dimensions,[24] presenting the dimension with the largest number of studies within each specialty.

### *Emergency medicine*
The seven studies in emergency medicine variously compared clinical care offered by PAs and physicians of various grades[34–37] and operational/service measures.[32 33] In only two of these studies was the comparison of PAs and other physicians in a system where the PAs were described as working 'solo', substituting for physicians at particular times of the day[32] or seeing patients without the input of the attending physician.[36]

Waiting or access outcomes were reported in one Canadian study[33]; the outcomes were leaving without being seen and waiting times. The presence of a PA was reported as significantly reducing the likelihood of a patient leaving without being seen by 44% (95% CI 31% to 63%, p<0.01), the crude rate being 6.5 without and 4.9% with a PA. The odds of a patient being seen within their benchmark wait time was 1.6 times greater (95% CI 1.3 to 2.1, p<0.05) when the PA was involved in the patient's care, with these analyses strengthened by adjustment for hospital, time of patient visit and acuity level.[33] However, the PA was an additional staff resource rather than a substitute in this study, giving extra coverage at the busiest times, alongside also newly appointed nurse practitioners, who increased the odds of being seen on target more than the PAs did, with an OR of 2.1.

Length of stay was considered in two studies,[32 33] with contradictory results in the comparison against physicians, from different interventions in terms of PAs. Arnopolin and Smithline[32] reported experienced ED PAs and physicians working solo at different times of day in a satellite unit. This study provided a direct comparison (and control for patient age in the analysis), with a result of a statistically significantly mean longer length of visit (8 min) for patients of PAs (82 min vs the physicians' 75 min, 95% CI –10 to –6, p<0.001), but also noted that differences in length of visit varied by diagnostic group, with PAs' patients between 5 and 32 min longer. In contrast, Ducharme et al[33] reported that where PAs were an additional staff resource alternating with nurse practitioners, PAs reduced their length of stay by 30% (mean 80 min reduction, 183 min vs 262 min, 95% CI 21.6% to 39%, p<0.01).

Cost was considered through total charge (hospital and physician charge) for the visit,[32] with a small but statistically significant decrease per patient reported when patients were treated by a PA, with differences (not statistically significant) by diagnostic groups.

Treatments offered, in terms of analgesia prescribing, were reported in three studies,[34 35 37] with conflicting findings. Secondary analysis of national (USA) ED survey data (1995–2004) reported no significant difference by type of provider in frequency of prescribing narcotic or non-narcotic analgesics and in the mean number of prescriptions per visit, but did observe a statistically significantly higher proportion of PAs' cases receiving a prescription compared with those of physicians and nurse practitioners (PAs 77.9%, physicians 75.5%, nurse practitioners 75.4%, p=0.001).[34] No adjustment for potential confounders was made. Using the same national survey data but for a subset for long bone fractures, secondary analysis for 1998–2003 reported similarly, with those seen by a PA having adjusted odds of 2.05 for receiving opiate analgesia in the ED (95% CI 1.24 to 3.29).[37] This well-powered retrospective cohort study of high quality differs from another study of similar quality with somewhat contrasting findings[35] in which for patients contacted at an undefined time (average 3 days following their ED visit) those attended by an emergency physician had adjusted odds of 3.58 (95% CI 2.05 to 6.24) for receiving pain medication while in the ED (29% of their patients) compared with those attended by PAs (10% of their patients), in a prospective cohort study based on patient self-report.[35] Although the period of time for this study is not specified, it first reported in 1998, perhaps suggesting the same decade of data was involved. These three studies did not report the PAs' place in the team or whether they added to or substituted for members of the medical team, nor whether they saw patients as part of a team or solo.

Two studies considered clinical outcomes of care. One, the oldest study in the review,[38] from 1995, reported that in a large sample of patients presenting with lacerations at the ED and seen by PAs there was no statistically significant difference in wound infection rates compared with

**Table 2** Main findings of included studies

| Specialty | Outcome measures | Finding(s) | Quality score | Key limitations | Study details |
|---|---|---|---|---|---|
| Emergency medicine | Length of visit (LOV) | Small but clinically insignificant differences (regression coefficient –8); LOV was 8 min longer when patients were treated by a PA (mean 82 min) than a physician (mean 75 min) (95% CI –10 to –6, p<0.001), although difference ranged from 5 to 32 min dependent on patient condition | 82% | ▶ Not randomised<br>▶ Differences by patient condition not explained<br>▶ Limited control for confounders | Arnopolin and Smithline[32] |
| | Total charge | Mean total charge was $159 when patients were treated by a PA and $164 by a physician (95% CI 2 to 14, p=0.013), regression coefficient –8 | | | |
| Emergency medicine | Leaving without being seen | Absolute improvement (not controlling for hospital or acuity) from 6.5% to 4.9%; when a PA was on duty, the likelihood that a patient left without being seen was less than half (44% (95% CI 31% to 63%), p<0.01), controlling for hospital and patient acuity | 73% | ▶ 2 months' data<br>▶ Sample size unclear | Ducharme et al[33] |
| | Wait time (triage to initial assessment) | When a PA was involved in patient care, the odds of the patient being seen within the benchmark wait time was 1.6 times greater than when the PA was not involved (95% CI 1.3 to 2.1), p<0.05, adjusting for hospital, acuity and time of day | | | |
| | LOS in ED | When a PA was involved in patient care, the LOS in the ED was shorter (mean: 262.4 min vs 182.9 min) than when a PA was not present (30.3%; 95% CI 21.6% to 39%), p<0.01 | | | |
| Emergency medicine | Proportion of visits in which medications are prescribed | Significant differences were observed between PAs if compared with physicians and to NPs in the proportion of visits in which medication was prescribed: PAs 77.9%, physicians 75.5%, nurse practitioners 75.4% (p=0.001) | 73% | ▶ Secondary data analysis<br>▶ No adjustment<br>▶ Treatment outcomes/ appropriateness not assessed | Hooker et al[34] |
| | Mean number of prescriptions written per visit | There were no significant differences among the three providers in mean number of prescriptions per visit (PA and physician 1.7, nurse practitioner 1.6). | | | |
| | Non-narcotic analgesic prescriptions | There were no significant differences among the three providers in the frequency of prescribing non-narcotic analgesics (p=0.16). | | | |
| | Narcotic analgesic/ NSAID prescription by type of provider | There were no significant differences among the three prescribers in the frequency of narcotic analgesics or NSAIDs recorded (p=0.15 and p=0.06, respectively). | | | |
| Emergency medicine | Analgesia prescribing | Emergency physicians gave some form of ED analgesia to 29% of patients, as compared with 10% of patients seen by PAs (OR 3.58; 95% CI 2.05 to 6.24), adjusting for sex, reported degree of pain and fracture. | 92% | ▶ Dependent on patient recall | Kozlowski et al[35] |

Continued

**Table 2** Continued

| Specialty | Outcome measures | Finding(s) | Quality score | Key limitations | Study details |
|---|---|---|---|---|---|
| Emergency medicine | 72-hour revisits to the ED | Patients treated only by PAs had significantly lower return rates (6.8%) than for the PA/emergency physician combined group (9.3%) and the emergency physician only group (8.0%), p=0.03. | 77% | ▲ No adjustment for significant differences in patient age, admission rate or patient complexity | Pavlik et al[36] |
| Emergency medicine | Proportion of patients with long bone fracture receiving analgesia | Patients seen by PAs had more than twice the odds of receiving opiates/narcotics (OR 2.05%; 95% CI 1.24 to 3.29) and were more likely to receive other analgesics (OR 1.72%; 95% CI 0.94 to 3.17) compared with those not seen by PAs | 100% | ▲ Changes in workload and documentation could have confounded results | Ritsema et al[37] |
| Emergency medicine | Patient wound infection rate | There were no significant differences in wound infection rates by practitioner level of training (medical students, 0/60 (0%); all residents, 17/547 (3.1%); physician assistants, 11/305 (3.6%); and attending physicians, 14/251 (5.6%); p=0.14). | 67% | ▲ Hawthorne effect<br>▲ Differences in wounds not controlled for | Singer et al[38] |
| Trauma and orthopaedics | Triage time to time seen by orthopaedic service (emergency department) (min) | PA presence resulted in a 205 min faster orthopaedic service response time (366 min vs 571 min; p=0.0006). | 91% | ▲ Exact cost savings difficult to determine<br>▲ Did not have a way of calculating savings for the time it took for patients to reach the OR from the time of triage<br>▲ Single site with two PAs | Althausen et al[39] |
| | Triage time to time of surgery (ER) (min) | PA presence resulted in a 360 min improvement in time to surgery (1139 min vs 1499 min; p=0.03). | | | |
| | Operating room complication rates (%) | There was no significant difference in the proportion of operating room complications with or without PAs (both 0.65%; p=0.9972). | | | |
| | The use of deep vein thrombosis prophylaxis (%) | The use of deep vein thrombosis prophylaxis increased by a mean of 6.73 percentage points (60.69% vs 53.96%; p=0.0084) with PA presence. | | | |
| | Postoperative antibiotic administration (%) | Postoperative antibiotic administration increased by 2.88 percentage points with PA presence (94.35% vs 91.47%; p=0.0302). | | | |
| | Postoperative complications (%) | There was a 4.67 percentage points decrease in postoperative complications with PA presence (8.16% vs 12.83%; p=0.0034). | | | |
| | Triage time to out of emergency department (min) | There was a 176 min decrease in total ER time with PA presence (270 min vs 446 min; p<0.001). | | | |

Continued

**Table 2** Continued

| Specialty | Outcome measures | Finding(s) | Quality score | Key limitations | Study details |
|---|---|---|---|---|---|
| | Operating room set-up time (min) | There was a marginally improved operating room set-up time by 0.43 min with PA presence (26.6 min vs 24 min; p=0.0034). | | | |
| | Time from wound closure to wheels out (operating room) (min) | There was no significant difference for this outcome when the PA was present (7.8 min vs 7.6 min; p=0.5914). | | | |
| | Average operating room time (min) | There was no significant difference in the average operating room time when the PA was present (70 min vs 74 min; p=0.44). | | | |
| | Cost savings (emergency department) ($) | Based on 50% collection of PA charges and emergency department time savings, per orthopaedic trauma patient seen, PAs saved the hospital $133.53 per patient, resulting in $41 394 in 1 year (310 patients). | | | |
| | Cost savings (operating room) ($) | The presence of a PA in the operating room resulted in savings of $3207 based on operating room costs (only set-up time was decreased with presence of the PA). | | | |
| | Hospital length of stay (days) | There was no significant difference in the hospital LOS when the PA was present if compared with the presence and the absence of PAs (7.96 days vs 8.57 days; p=0.2662). | | | |
| Trauma and orthopaedics | Patient satisfaction | 91.3% of hip patients (total=626, 58.5% response) reported being satisfied or very satisfied and 87.7% of knee patients reported being satisfied or very satisfied with PAs at 1-year follow-up (after surgery) | 32% | ▲ Methods are not fully described, for example, no description of data analysis<br>▲ Sample is not described<br>▲ Is this a study about PAs or about the two-room operating model?<br>▲ Patient satisfaction with the surgery at 1 year cannot be attributed to the PA | Bohm et al[40] |

Continued

**Table 2** Continued

| Specialty | Outcome measures | Finding(s) | Quality score | Key limitations | Study details |
|---|---|---|---|---|---|
| | Perceptions of healthcare providers and patients about PAs | Patients: overall patients expressed very positive opinions of PAs who were helpful in providing information and explaining aspects of their care. Ward nurses: felt that patient care, information flow and patient rounds were enhanced by the PAs; ambiguous as to whether PA tasks fell within the scope of nursing. Orthopaedic surgeons: overall the surgeons had very positive opinions of PAs—100% agreement with all survey items: 'a fully trained PA provides surgical assistance equal to an R5 (fifth year of a residency programme)'; 'the presence of PA has improved your job satisfaction'; 'the presence of a PA has safely allowed you to do more surgical volume'; 'the care of your patients in the OR is improved by the assistance of PAs'; 'PAs greatly decrease the amount of 'scut work' that you have to do'. Operating room nurses: overall OR nurses reported that PAs were valuable team members; improved the care of orthopaedic surgery patients in the operating room; provided surgical assistance superior to family practitioners; and were necessary to run two operating rooms. Orthopaedic residents: nearly unanimous that PAs reduced their workload and they generally felt that PAs relieved them of clinical responsibilities so that they could attend teaching. | | | |
| | Costs | The cost of employing three PAs in 2006 (between $270 000 and $327 000) was found to be similar to the foregone general practitioner (GP) surgical assist fees of $270 226.88. | | | |
| | Time savings | PAs were found to 'free up' 204 hours/year (the equivalent of four 50 hours' work weeks) for their supervising physician (p=notreported). Furthermore, they potentially freed GPs from the operating room to spend more time delivering primary care. | | | |
| | Throughput | Increased the volume from three to seven primary joint surgeries per day through the use of double rooms in 2006. | | | |
| | Waiting time | Median wait time for surgery decreased from 44 to 30 weeks. | | | |

Continued

**Table 2** Continued

| Specialty | Outcome measures | Finding(s) | Quality score | Key limitations | Study details |
|---|---|---|---|---|---|
| Trauma and orthopaedics | Fracture malunion (maximum angulation criteria) at last clinic visit | Likelihood of malunion did not differ significantly if the providers included a PA or not (28% vs 56%, Fisher's exact test p=0.13) or by number of PAs (p=0.11). | 82% | ▲ Unadjusted comparisons<br>▲ Difficult to assess how much of the care was carried out by PAs (analysis is cases with any PA involvement vs cases with no PA involvement) | Garrison et al[41] |
| Trauma and orthopaedics | ▲ Perceptions and experiences with the PA | ▲ Preoperative care: PA triages, conducts most activities without direct supervision.<br>▲ Operating room: PAs' integration into the OR went well; staff appreciate consistency of the PA; PA acquired skills in a graduated manner—now 'preps and closes with patients in OR'.<br>▲ Postoperative care: takes on some of surgical extender role but the role is missed after hours; PA sees 60%–70% of all inpatients, freeing up the surgeon; full integration limited by needs for cosignature and verification of orders.<br>▲ Follow-up outpatient care: clinic flow improved.<br>▲ PA is a collaborative member of the team (most mean ratings >4 out of 5. | 55% | ▲ Unable to ascertain which data are descriptive quantitative or gained from qualitative interviews | Hepp et al[42] |
| | ▲ Patient rating of quality of care | All patients responded positively to the PA role; overall rating of PA care of 9.65 of 10. | | | |
| | ▲ Expected and actual operating room times | Double-room experiment: actual preparation time 39% longer than expected and postsurgery time 37% less than expected (absolute times not given) surgeon time 21% less; 2 hours/day saving | | | |
| | Total new patients seen | Preoperative care: 30% increase in numbers of patients seen, noticed in the first year | | | |

Continued

**Table 2** Continued

| Specialty | Outcome measures | Finding(s) | Quality score | Key limitations | Study details |
|---|---|---|---|---|---|
| Trauma and orthopaedics | Overall mortality | The introduction of PAs to the core trauma panel (group 3 vs group 2) decreased overall mortality (2.80% vs 3.76%, adjusted OR 0.74 (95% CI 0.55 to 0.99), p=0.05). Furthermore, the introduction of PAs to general surgery residents (group 3 vs group 1) decreased overall mortality (2.32% vs 3.82%, adjusted OR 0.6 (95% CI 0.45 to 0.81), p=0.003). | 100% | ▲ Not all the covariates which could be significantly associated with outcomes were collected (eg, changes in care). <br> ▲ The group 1 period was characterised by a transition from on-call attending surgeons to in-house surgeons and the outcomes may not be homogenous across the study period. <br> ▲ Other changes were made, not just individual staff type. | Mains et al[43] |
| | Hospital LOS | The introduction of PAs to the core trauma panel (group 3 vs group 2) reduced mean and median hospital LOS (4.32 days vs 4.69 days, p=0.05; and 3.74 days vs 3.88 days, p=0.02, respectively). As well, the introduction of PAs to general surgery residents (group 3 vs group 1) reduced mean and median hospital LOS (4.32 days vs 4.62 days, p=0.05; and 3.74 days vs 3.94 days, p=0.003, respectively). | | | |
| Trauma and orthopaedics | Collaborative relationship | Participation during trauma alert calls: PA 100%; resident 51% overall, 88% during on-duty hours. Involvement in minor procedures PA 100% when residents off-duty, 91% overall; resident 95% during on-duty hours, 83% overall. | 82% | ▲ Investigators not blinded and all work in the trauma centre investigated <br> ▲ No sample size calculation <br> ▲ Single site with two PAs <br> ▲ Minimal description of data collection method | Oswanski et al[44] |
| | Transfer time | After controlling for age, gender, race and severity of illness, there was no significant difference in the mean transfer rate overall or for any subpopulation (destination) between years 1998 and 1999. | | | |
| | LOS | After controlling for age, gender, race and severity of injury, there was no significant difference in the mean LOS overall between years 1998 and 1999. | | | |
| | Mortality rate | Mortality rate for all patients admitted to the trauma service was 2.2% for both 1998 (8/293) and 1999 (13/479). | | | |

Continued

**Table 2** Continued

| Specialty | Outcome measures | Finding(s) | Quality score | Key limitations | Study details |
|---|---|---|---|---|---|
| Internal medicine | 30-day all-cause readmission | No statistically significant difference in odds of readmission between expanded PA (14%) and conventional PA (13.7%) groups (OR 0.95; 95% CI 0.87 to 1.04; p=0.27) | 91% | ▲ Non-randomised patient allocation<br>▲ Use of secondary data<br>▲ Readmission to the same hospital only | Capstack et al[45] |
| | Inpatient mortality | No statistically significant difference in odds of mortality between expanded PA (1.3%) and conventional PA (0.99%) groups (OR 0.89; 95% CI 0.66 to 1.19; p=0.42) | | | |
| | Cost of care | Statistically significant difference in mean patient charge between expanded PA ($7822) and conventional PA ($7755) groups (3.52% lower; 95% CI 2.66% to 4.39%; p<0.001) | | | |
| | Consultant use | No statistically significant difference in utilisation of consultants between expanded PA (1.3%) and conventional PA (0.99%) groups (OR 1.0; 95% CI 0.94 to 1.07; p=0.90) | | | |
| | Length of stay | No statistically significant difference in length of stay between expanded PA (4.1±3.9 days) and conventional PA (4.3±5.6 days) groups (effect size, 0.99 days shorter; 95% CI 0.97 to 1.01 days; p=0.90) | | | |
| Internal medicine | Relative value units (RVU; ie, costs) | (1) Radiology RVUs: there were no statistically significant differences between PAs and residents; (2) total RVUs (excluding pharmacy data): PAs used significantly fewer resources when compared with resident services for pneumonia care (p=0.004), although had a higher mortality rate (% and p value not reported). For all other diagnoses there were no statistically significant differences in total RVUs between PAs and residents; (3) laboratory RVUs: there were statistically significant differences between PAs and residents in laboratory relative value units for stroke (p=0.015), pneumonia (p=0.003) and CHF (p=0.004). In each case, PAs' RVUs were lower than those of residents. | 86% | ▲ RVU figures are not explained<br>▲ Non-random group assignment<br>▲ Single centre | Van Rhee et al[46] |
| | Length of stay (LOS) | There were no significant differences in LOS between PAs and residents after adjusting for admitting physician effect and other covariates. | | | |

Continued

**Table 2** Continued

| Specialty | Outcome measures | Finding(s) | Quality score | Key limitations | Study details |
|---|---|---|---|---|---|
| Mental health | Perceived effect and challenges of delivering psychiatric care with the PA model | Participants described: improved access to primary care for patients; more timely access to psychiatric appointments and longer appointments; equal team cohesion for the PA or the psychiatrist; decreased wait times and improved access to tertiary care and screening programmes; and implementation challenges of triage hierarchy and patient understanding of the term physician assistant | 45% | ▲ Qualitative analysis methods described without detail <br> ▲ Short report with overview of themes; no quotations | McCutchen et al[47] |

CHF, congestive heart failure; ED, emergency department; ER, emergency room; GP, general practitioner; NP, nurse practitioner; NSAID, non-steroidal anti-inflammatory drug; PA, physician assistant/associate.

other medical staff providers (medical students, residents and attending physicians).[38] However, the authors noted a potential Hawthorne effect as all wounds had been evaluated by an attending physician prior to allocation to one of the medical team members, based on their level of training. It was noted that PAs in this study, with 9–12 years' experience, were classified as experienced (not junior) practitioners. The other, newer, study[36] used a proxy measure of clinical safety, that is, the 72 hours' reattendance (recidivism) rate to the ED for children aged 6 and younger, and reports that this was significantly lower for those patients treated only by a PA (6.8% vs emergency physician 8.0%, p=0.03), in a large study. However, these rates were unadjusted, and the characteristics of the study population show statistically significantly different mean ages and rate of admission in the patients treated in each group, with PAs seeing the older of the children who were much less likely to be admitted. Although analysis of the recidivism rates by Emergency Severity Index score for patients seen by PAs versus doctors found no statistically significant differences between groups and the authors conclude that PA providers deliver comparable care, the authors themselves consider that it is not known if PAs would have made the same decisions as physicians for the same group of patients.

### Trauma and orthopaedics

Six papers reported on PAs working in trauma and orthopaedics. These spanned a 14-year period. Four[39 41 43 44] focused on an aspect of provision of a hospital trauma service; and two considered planned inpatient care.[38 42]

Three studies described how PAs were substituting for doctors, for residents[44] or surgical assistants,[40 42] while the others presented service reorganisations of which PAs were a part, seemingly an addition to the pre-existing medical team.[39 41 43] The outcomes assessed were numerous—patient satisfaction, perceptions of other clinical staff, costs, time of various aspects of care, patient throughput, length of stay, fracture malunion and operative complications and mortality. The strength of evidence for each outcome is now assessed.

Two prospective studies of the addition of PAs to surgical teams, preoperatively, intraoperatively and postoperatively,[40 42] reported both patient satisfaction and acceptability of PAs to other clinical staff from surveys of these groups. Positive results were presented from both studies' patient satisfaction surveys, in large[40] and small[42] response numbers, reporting 91.3% of hip and 87.7% of knee patients being satisfied or very satisfied[40] and an overall rating of PA care of 9.65 out of 10[42] although no comparator data were collected. The reports of staff were more mixed by staff group in Bohm et al's study[40] with physician team members being positive (100% agreement with all survey items on the positive contribution of PAs) and nursing staff more equivocal, expressing concern about the overlap of tasks traditionally considered to be the responsibility of nurses; and by impact in different parts of the surgical journey in Hepp et al's[42]

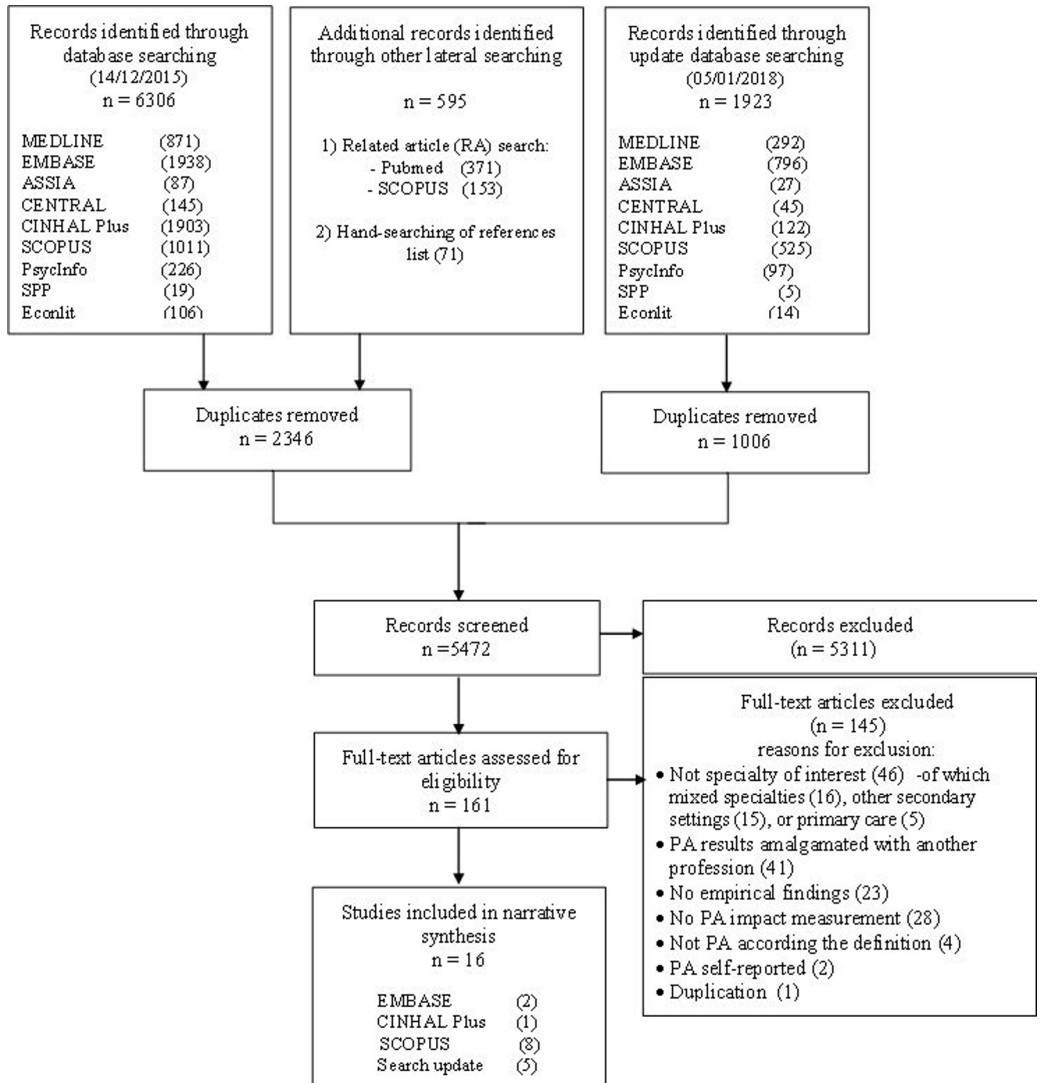

**Figure 2** Preferred Reporting Items for Systematic Reviews and Meta-Analyses (PRISMA) flow chart. PA, physician assistant/associate.

study, where staff ratings were mostly above 4 out of 5, agreeing or strongly agreeing that the PA was a collaborative team member. Staff appreciated continuity and PA advances in skills in the operating room, but did not feel the role could offer everything a previous surgical extender did postoperatively, despite being a collaborative team member.[42]

Operational measures were addressed in five of the studies in this specialty, split into a number of outcomes pertaining to time[39 40 42–44] and to cost.[39 40]

The evidence of the impact of PAs on access times was equivocal. One study reported how the wait to be seen by the orthopaedic service in the ED section of their orthopaedic pathway was significantly shortened (366 min vs 571 min; p=0.0006) when PAs were substituted directly for doctors, although the authors attributed this to a combination of factors, and not just to the PAs, including more registered nurse cover, introduction of a family practice resident and other changing practices.[39] Another found the same when PAs were added to the team as part of

larger trauma team reorganisation.[39] Median number of weeks to wait for surgical procedures was also reported to be reduced from 44 to 30 weeks,[40] attributed by the authors to the use of two operating theatres by the surgeon, made possible by the PA preparing and finishing the case, similarly to the 30% increased throughput in the number of new patients in the preoperative stage.[42]

In terms of time, two studies[39 42] reported in detail on operating room times—set-up, wound closure to out of theatre, average operating room time and postsurgery time. Althausen et al[39] only noted a minimal (not statistically significant—26.6 min vs 24 min; p=0.0034) difference for set-up time in a direct comparison study, while Hepp et al[42] describe a 39% reduction in time at this stage. PAs also released time for supervising physicians—204 hours/year (p=not reported)[40] or 2 hours/day,[42] and for GPs (not quantified), who had previously acted as surgical assistants.[40] Three high-quality studies[39 43 44] reported variably on length of hospital stay, with one showing a significant reduction (3–4 hours, a

fraction of 1 day) for all patients when PAs were an addition to either the resident physician team (mean 4.32 days vs 4.62 days, p=0.05; and median 3.74 days vs 3.94 days, p=0.003) or reorganised trauma panel (mean 4.32 days vs 4.69 days, p=0.05; and median 3.74 days vs 3.88 days, p=0.02)[43] and two replacement studies finding no difference—when carrying out adjusted analyses of 1 year against another[44] or when PAs were present or not.[39]

Evidence regarding cost was again mixed. Bohm et al[40] suggest the actual costs of employment for three PAs (between $270 000 and $327 000) were similar to those of the GPs they replaced ($270 226.88) in the operating room but argue an opportunity cost for others through released time for the supervising physicians. However, a non-replacement model, Althausen et al[39] reported specific cost savings in the ED ($133.53 savings per patient, $41 394 in 1 year) and operating room ($3207 savings) based on time reduction and PA charges (taking account that only 50% of PA costs were covered through charges).

As well as these operational measures, these studies also reported health outcomes, and all reported no difference[41] or improvement in these.[39 43 44] Two considered the rate of complication from procedures involving PAs[37 41] and two reported on mortality.[43 44] In terms of operating room complication rates[39] or the likelihood of fracture malunion if the providers included a PA,[41] these did not differ significantly from those of other providers, but postoperative complications were reported to have decreased (8.16% vs 12.83%, p=0.0034) and antibiotic use (94.35% vs 91.47%, p=0.0302) and deep vein thrombosis prophylaxis (60.69% vs 53.96%, p=0.0084) increased (statistically significantly) for cases with a PA present (although it is noted that the tables in this paper presented findings contradictory to the text and abstract).[39] One study assessing mortality in two year-long periods reported that involvement of PAs in the clinical team had no effect on overall mortality rates[44] while another found that mortality decreased by approximately 1% with the introduction of PAs to a trauma panel (9.67% vs 12.21%, adjusted OR 0.77; 95% CI 0.55 to 0.99, p=0.13) and 1.5% to general surgery residents' teams (9.03% vs 14.83%, adjusted OR 0.6; 95% CI 0.41 to 0.80, p=0.003).[43] However, this could not be directly attributable to the addition of the PA because contemporaneous improvements in efficiency of the trauma service occurred.

### Acute internal medicine

The two studies considering PAs in acute internal medicine both examined resource use and clinical outcomes[45 46] in replacement studies, one prospectively examining the impact of PAs in place of interns/residents,[46] the other retrospectively comparing outcomes where PAs made up a greater or lesser proportion of the medical team staff, in place of physicians.[45] Both studies measured length of stay, direct costs and inpatient mortality for patients with diagnoses of cerebrovascular accident, pneumonia, acute myocardial infarction

discharged alive, congestive heart failure (CHF) and gastrointestinal haemorrhage,[46] and those with a principal medical (non-surgical, non-obstetrical) diagnosis code[45]; the latter study also measuring 30-day all-cause readmission. Neither study reported any significant differences in length of stay between groups, with length of stay considered to be a proxy for severity of illness. Cost in terms of relative value units (RVU, based on billing information for physician-ordered items, excluding administrative costs outside of the physician's control) was also mostly similar although laboratory RVUs were lower for PAs, that is, they ordered fewer investigations after adjustment for demographics in each diagnostic group (for stroke p=0.015, pneumonia p=0.003 and CHF p=0.004). In each case, PAs' RVUs were lower than those of residents.[46] Similarly, Capstack et al[45] reported a statistically significantly lower mean patient charge for the expanded PA group ($7822 vs $7755 for the conventional PA group (3.52% lower (95% CI 2.66% to 4.39%); p<0.001)). Inpatient mortality was stated to be higher for the PA group in pneumonia care only,[46] although the authors reported neither the percentage nor statistical values, and the larger study reported no significant differences in mortality or 30-day all-cause readmission.[45] The authors concluded that PAs used resources as effectively as, or more effectively than, residents[46] at the same time as providing similar clinical quality.[45]

## DISCUSSION
### Principal findings
This systematic review identified a large number of studies of PAs working in secondary care settings, internationally. However, once studies were excluded that did not meet the inclusion criteria, only 16 papers remained. Most of the included studies were from the emergency medicine and trauma and orthopaedics specialties, with two from acute internal medicine and one from mental health. We found no studies in our other specialty of interest—care of the elderly—where another larger grouping of PAs worked in the UK according to a national survey[18] at the time of planning this review. Several of the studies were of high quality, providing comparative data, and some contained statistical adjustments to address confounding; however, all findings were observational. While we recognise that trials are rarely feasible in this type of workforce intervention, adjustment for confounding by indication is a serious challenge in this setting, especially when using a limited routine data source, and residual confounding from imperfect measures of severity[48] and bias from adjusting for covariates that were not confounders[49] were likely. Quality also varied widely. This is noteworthy considering that this was a relatively recent set of papers. In addition, comparison and synthesis has been limited by the mix in the papers of those who measure outcomes where PAs are an addition to a team (presenting difficulties in attributing the outcomes to PAs as opposed to any other increase in team capacity) and those where PAs

substitute for other physicians where the contribution of PAs themselves is actually being measured. Although every paper reported the contribution of PAs in its specialty/subspecialty as overall positive, it is important that the following summary of the main findings of the review is considered in the context of the issues of method and methodological quality.

Results were spread across a number of outcomes, though those related to operational measures—waiting times or times taken for treatment, as well as patient satisfaction—were most prevalent. Outcomes reported when employing PAs in emergency medicine were varied. Operational performance results reported were decreased waiting time and reduced length of stay in the ED,[33] an increase in length of visit for those seen by PAs[32] and reduced charges.[32] Healthcare outcomes reported were no difference in 72 hours' revisits to the ED[36] or wound infection rate,[39] and differences which were difficult to interpret, for example, an increased prescription rate,[34] or increase[37] or decrease in analgesia prescribing.[35] The messages are remarkably similar for trauma and orthopaedics. Operational measures highlighted no difference to[44] or reduced[39 40 42 43] waiting times in the emergency, operative and postoperative phases of care; released physician time[40 42] and reduced cost.[39] Here the evidence on health outcomes was mostly positive—increased adherence to treatment processes such as antibiotic administration,[39] reduced postoperative complications,[39] no difference in fracture malunion[41] and either no difference[44] or a reduction[43] in mortality. High patient satisfaction and staff acceptability, although with some caveats, were also reported.[40 42]

The two studies in internal (acute) medicine were of high quality and were among the few replacing physicians with PAs. Both found no differences in clinical outcomes between PAs and residents, or in length of stay, although lower costs were reported.[45 46] In mental health, the one study's qualitative evidence points also to acceptability of the role through team cohesion and improvements in whole system working.[47]

Summarising across the specialties we have reported five studies where PAs were an addition to the team.[33 39 42 43 46 47] In these more patients are reported to have been treated; waiting, ED and operating room times are said to have been shorter and mortality to be lower; however, assessment of the contribution of PAs as opposed to any increase in team capacity is limited. Eight studies which compared outcomes of care by PAs and physicians either when one or the other was providing care or when PAs were substituting overall for physicians[32 35 36 38 40 44–47] presented mixed results: either no or a very small difference to length of stay, reduced resource used but at equal or reduced cost, some time savings to senior physicians, lower analgesia prescribing, no difference in wound infection rate, inpatient mortality or reattendance, or in acceptability to staff and patients. In three of the studies we do not know if the PAs were additions or substitutions but two reported

higher prescribing by PAs.[34 37] and one no difference in negative outcomes from fracture.[41]

## Strengths and weaknesses

This review has systematically assessed the body of PA literature most immediately applicable to the current UK secondary care setting. We selected the five specialties in which PAs in the UK were mostly reported to be working[18] and therefore drew together the evidence of most relevance in that context and noted prominent gaps in evidence. However, this excluded evidence from other specialties. We excluded any studies including intensive care data as this overlapped with acute medicine in many abstracts and we could not separately draw this out, and similarly we excluded studies with medical and surgical specialties combined. We note that this literature appeared to include a greater proportion of studies with stronger study designs, including prospective and randomised designs; in particular we have excluded the recent matched controlled large studies from the Netherlands in which several specialties—some within and some without our inclusion criteria—were studied.[50 51]

All of the included papers were from North America, with the majority from the USA, where health service organisation and the PA role may differ from that in other countries developing the PA role. In the USA, PAs can prescribe and order ionising radiation, and are, as a body, more experienced than in countries more recently embracing this role.

We planned to carry out meta-analysis as appropriate to the literature included. The diversity of intervention as in initiation of PAs or change to PA practice being measured prevented this, as did identifying the effect of PAs when there were other simultaneous changes, even where a body of literature pertaining to a particular outcome measure, such as length of stay, was included. Although narrative review is more limited in its precision, in following a framework for this, we have aimed to provide a clear rationale for the synthesis and conclusions we draw from it.

## Meaning of the study

This evidence is heavily weighted towards process times and patient satisfaction, with much less on health outcomes, although outcomes are crucial to assess safety of practice for all clinicians. Similar findings have been reported in a systematic review of new (non-medical) roles in emergency medicine—reductions in waiting times in EDs, high level of patient satisfaction, confidence and acceptance of the roles.[52] Evidence also suggests that the perception of waiting times and satisfaction are correlated.[53]

Evidence from outside of the USA is very slim, as is evidence from multicentre studies. The implications of this for policy can be seen in two ways.

First, the limitations to evidence could be considered a cause for some concern, particularly in light of exponential growth in training numbers for PAs in England (alongside other UK countries),[54] government support for increased

numbers (in primary care at least)[10] and for recent consultation on the introduction of statutory regulation for PAs, alongside judgement by employers and workforce planners of the role's value, alongside other medical associate professions.[55 56] Numbers of PAs are also rising rapidly in the USA.[4] That said, the evidence presented in this review is positive and likely supportive of the direction of travel in policy. In addition, the case for PAs in the UK secondary care setting is made on the stability they might offer to medical teams and their broad knowledge in the face of hyperspecialisation[57] and recently acquired knowledge—although not covered in this review due to its inclusion of PAs from across multiple specialties—suggests that PAs in England work in teams of multiple medical and other clinical staff grades[58] and that they are seen primarily as a resource where there are significant medical staffing issues.[59] High-quality, multicentre-matched controlled substitution evidence from the Netherlands[50 51] reassuringly also offers similar evidence to that included in our review regarding no difference in a large number of inpatient and postdischarge clinical outcomes, alongside an increase in patient satisfaction. The study found no difference in total healthcare costs or quality-adjusted life years, despite lower personnel costs. The authors conclude that PA substitution appeared safe. The studies included in this review can be seen as complex interventions in complex systems and yet this has not been considered in the conclusions the authors draw. Well-controlled studies are needed to fill in the gaps in our knowledge about the outcomes of PAs' contribution to the secondary care. More such evidence is required as well as further evaluation from a realist perspective—considering context, mechanisms and outcome—if PAs cannot be separated from service; measurement would use the principles of realist complex intervention science[60] or process evaluation to 'Clearly describe the intervention and clarify causal assumptions (in relation to how it will be implemented, and the mechanisms through which it will produce change, in a specific context).'[61]

## CONCLUSION

Modest research evidence exists on PAs working in emergency medicine, trauma and orthopaedics, and acute internal medicine; very limited evidence in mental health and none meeting our criteria in care of the elderly. The focus of the research is mainly on organisational and financial implications because increasing throughput of patients, while containing costs and without adversely affecting outcomes, is fundamental to the rationale for the PA role. Evidence shows that use of PAs can achieve this objective. The PAs worked as additions as well as substitutes in complex systems where work is organised in teams which creates challenges for identifying cause and effect. PA employment is also often part of wider service redesign or staffing changes in response to other changes, for example, availability of medical staff. The evidence here suggests that PAs can make a positive contribution to medical care and medical teams. Further research to the standard of more recent publications is needed to elucidate the impact of PAs in different specialty areas, including comparators, and reporting on more than one setting, including countries in which the PA role is expanding rapidly.

**Contributors** MH led the design, execution and writing of this paper, under the direction of the study's PI VMD and supported by discussion with and written feedback from all coauthors (CW, FP, HG, SL, JP, RG, JG, LN) on the design of the review and interpretation of findings. In addition, MH, CW, FP and VMD searched for literature and carried out data extraction and quality assessment. All authors (CW, FP, HG, SL, JP, RG, JG, LN, VMD) contributed intellectual content to the paper.

**Funding** This project was funded by the National Institute for Health Research Health Services and Delivery Research Programme (project number 14/19/26).

**Disclaimer** The views and opinions expressed herein are those of the authors and do not necessarily reflect those of the HS&DR Programme, NIHR, NHS or the Department of Health.

**Competing interests** SL: University of Surrey runs a Physician Associate course. JP: chairs the UK and Ireland Board for Physician Associate Education and is the director of the Physician Associate programme at the University of Birmingham.

**Patient consent** Not required.

**Provenance and peer review** Not commissioned; externally peer reviewed.

**Data sharing statement** No additional data are available.

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
