## [Reviewer comments · BMJ Open]

ARTICLE DETAILS

TITLE (PROVISIONAL)	The contribution of physician assistants/associates to secondary care: a systematic review
AUTHORS	Halter, Mary; Wheeler, Carly; Pelone, Ferruccio; Gage, Heather; de Lusignan, Simon; Parle, Jim; Grant, Robert; Gabe, Jonathan; Nice, Laura; Drennan, Vari

VERSION 1 – REVIEW

REVIEWER	Tamara S. Ritsema George Washington University, USA St. George's, University of London, UK
REVIEW RETURNED	13-Oct-2017

GENERAL COMMENTS	1) Overall well designed study which nicely lays out the literature for 4 specialties within secondary care. Rigorous evaluation criteria are used by the authors for their assessment. Assessment and conclusions are fair - the authors acknowledge the difficulty of performing an evaluation of a new health profession added to an existing team. 2) In the introduction, the authors state that there are roughly 93,000 PAs in the US. This data is from 2013 and is out of date, primarily because PA education in the US has expanded substantially in the last 10 years, and 8,000+ people are graduating as US PAs each year now. A better reference would be the National Commission on the Certification of Physician Assistants, which is the regulatory body for PAs. They say as of 31/12/16, there are 115,547 PAs. Here's the link: https://prodcmssstoragesa.blob.core.windows.net/uploads/files/2016StatisticalProfileofCertifiedPhysicianAssistants.pdf
---

REVIEWER	Rebecca Hoskins University Hospitals Bristol NHS Foundation Trust Emergency Department, Bristol Royal Infirmary Bristol BS2 8HW UK
REVIEW RETURNED	05-Dec-2017

GENERAL COMMENTS	Thank you for this useful and timely systematic review of the role of PAs internationally. I think this is a thorough and comprehensive systematic review. I was disappointed though not to see a consideration in the introduction (or discussion) in order to provide some context about the fact that PAs are not a registered professional body in the UK (yet) and as a consequence cannot request investigations containing ionising radiation or become independent prescribers. How does this impact on satisfaction and patient flow in the UK and did the studies not allude to this. Inclusion
---

VERSION 1 – AUTHOR RESPONSE

Editorial Team comments

- Please update the literature search, which is over 12 months old now

We have completed a full update, with searches carried out on 5th January 2018

As this update identified 917 new references on search and resulted in the inclusion of five additional papers, changes are throughout the document.

- The 'Strengths and Limitations' section on page 5 needs improving. It should be clearer why each bullet point is a strength or limitation and each point should relate to the study's design or methods.

We have altered the strengths and limitations to be clearer in their focus on design and methods (page5)

- Please justify the quality assessment tool selected and explain better how the tool works

We have added a brief explanation of why the tools were selected and reference the full description of their content (and the validation of that) (page 10)

- The abstract and the results section are very poor in numbers. Considering the tables include a reasonable number of statistics, perhaps you could include some in the abstract and results section?

We have included the statistics from the tables in the main text. Due to the number of different measures provided we respectfully suggest that the abstract would not be of the required length if statistical results were presented there. (pages throughout the findings section)

- You should discuss any policy implications of your findings. For instance, in the last couple of years, the UK has “imported” a significant number of physician associates to work in the NHS, in primary care:

<http://www.pulsetoday.co.uk/your-practice/practice-topics/employment/nhs-offering-50k-per-year-for-us-physician-associates-to-practise-in-underdoctored-areas/20010929.article> . Moreover, the UK is already investing heavily in the training of physician associates: <http://www.pulsetoday.co.uk/your-practice/practice-topics/education/nhs-to-spend-15m-on-training-1000-gp-physician-associates-by-2020/20033552.article> . Has the experience with PAs in secondary care influenced their increasing uptake in primary care?

We have increased the section in the discussion that already referenced the expansion in education places for PAs using the source HEE document, to be more explicit about the impact on policy. (page40)

Reviewer: 1

- 1) Overall well designed study which nicely lays out the literature for 4 specialties within secondary care. Rigorous evaluation criteria are used by the authors for their assessment. Assessment and conclusions are fair - the authors acknowledge the difficulty of performing an evaluation of a new health profession added to an existing team.

Thank you for your favourable view on this paper

- 2) In the introduction, the authors state that there are roughly 93,000 PAs in the US. This data is from 2013 and is out of date, primarily because PA education in the US has expanded substantially in the last 10 years, and 8,000+ people are graduating as US PAs each year now. A better reference would be the National Commission on the Certification of Physician Assistants, which is the regulatory body for PAs. They say as of 31/12/16, there are 115,547 PAs. Here's the link: <https://prodcmsstoragesa.blob.core.windows.net/uploads/files/2016StatisticalProfileofCertifiedPhysicianAssistants.pdf>

Thank you for this helpful up to date reference to the numbers of PA in the US. We have updated the number in the introduction and replaced our previous reference [4]. (page 6)

Reviewer: 2

- Thank you for this useful and timely systematic review of the role of PAs internationally. I think this is a thorough and comprehensive systematic review. I was disappointed though not to see a consideration in the introduction (or discussion) in order to provide some context about the fact that PAs are not a registered professional body in the UK (yet) and as a consequence cannot request investigations containing ionising radiation or become independent prescribers. How does this impact on satisfaction and patient flow in the UK and did the studies not allude to this. Inclusion of this issue would give a more balanced picture I believe
Thank you for your favourable review of this paper.

We have now added the issue of registration and its associated prescribing and ionising radiation issues in the introduction as part of our reference to the growth of the PA in primary care and the evidence related to that. (pages 6 and 40)

Other changes the author team has made

As a result of the update of the review we have made some changes to our conclusion – the message remains the same, but we acknowledge that the amount and quality of evidence has improved (page 52)

As Health Education England and the Department of Health are now routinely referring to this group of staff in the UK as 'physician associates', no longer 'physician assistants' but the internal literature remains 'physician assistant' we wonder if the reviewers and editor would consider a change in the paper's title to reflect both terms (pages 1 and 7).

We thank you for your consideration of this revised paper, and look forward to your response.

VERSION 2 – REVIEW

REVIEWER	Dr Rebecca Hoskins University of the West of England, UK
REVIEW RETURNED	02-Mar-2018
GENERAL COMMENTS	I have reviewed this paper before and feel the issues raised have been addressed